# Soluble MAC is primarily released from MAC-resistant bacteria that potently convert complement component C5

**Dennis J Doorduijn[1], Marie V Lukassen[2], Marije FL van 't Wout[1], Vojtech Franc[2], Maartje Ruyken[1], Bart W Bardoel[1], Albert JR Heck[2,3], Suzan HM Rooijakkers[1]\***

[1]Department of Medical Microbiology, University Medical Center Utrecht, Utrecht, Netherlands; [2]Biomolecular Mass Spectrometry and Proteomics, Bijvoet Center for Biomolecular Research and Utrecht Institute for Pharmaceutical Sciences, University of Utrecht, Utrecht, Netherlands; [3]Netherlands Proteomics Center, Utrecht, Netherlands

**Abstract** The membrane attack complex (MAC or C5b-9) is an important effector of the immune system to kill invading microbes. MAC formation is initiated when complement enzymes on the bacterial surface convert complement component C5 into C5b. Although the MAC is a membrane-inserted complex, soluble forms of MAC (sMAC), or terminal complement complex (TCC), are often detected in sera of patients suffering from infections. Consequently, sMAC has been proposed as a biomarker, but it remains unclear when and how it is formed during infections. Here, we studied mechanisms of MAC formation on different Gram-negative and Gram-positive bacteria and found that sMAC is primarily formed in human serum by bacteria resistant to MAC-dependent killing. Surprisingly, C5 was converted into C5b more potently by MAC-resistant compared to MAC-sensitive *Escherichia coli* strains. In addition, we found that MAC precursors are released from the surface of MAC-resistant bacteria during MAC assembly. Although release of MAC precursors from bacteria induced lysis of bystander human erythrocytes, serum regulators vitronectin (Vn) and clusterin (Clu) can prevent this. Combining size exclusion chromatography with mass spectrometry profiling, we show that sMAC released from bacteria in serum is a heterogeneous mixture of complexes composed of C5b-8, up to three copies of C9 and multiple copies of Vn and Clu. Altogether, our data provide molecular insight into how sMAC is generated during bacterial infections. This fundamental knowledge could form the basis for exploring the use of sMAC as biomarker.

## Editor's evaluation

This manuscript describes in detail the strategies employed by certain bacteria to defend against lytic attack by the membrane attack complex (MAC) of complement. The major new finding is that during complement activation, these MAC-resistant bacteria are able to process and release considerable amounts of C5a as well as large amounts of a soluble form of the MAC (C5b-9) that has less C9 than the active form that promotes bacterial cell lysis.

## Introduction

The complement system is a part of the human immune system that plays a crucial role in clearing invading bacteria to prevent infections. The complement system consists of soluble plasma proteins that circulate as inactive precursors (*Gasque, 2004*; *Ricklin et al., 2010*). When complement is activated at the bacterial surface, a proteolytic cascade is triggered that labels the surface with convertase

**\*For correspondence:**
s.h.m.rooijakkers@umcutrecht.nl

**Competing interest:** The authors declare that no competing interests exist.

enzymes (*Gasque, 2004*). These convertases initially convert C3 into anaphylatoxin C3a and C3b, labelling bacteria for phagocytosis by neutrophils (*Merle et al., 2015*). As terminal step in the pathway, these convertases also convert C5 into pro-inflammatory C5a, which recruits and activates neutrophils, and C5b. C5b, together with C6, C7, C8, and up to 18 copies of C9, assembles a large, ring-shaped membrane attack complex (MAC) pore (*Bhakdi and Tranum-Jensen, 1978*; *Müller-Eberhard, 1986*; *Menny et al., 2018*; *Doorduijn et al., 2019*). MAC pores can efficiently kill Gram-negative bacteria, although some serum-resistant Gram-negative bacteria can survive killing by MAC pores (*Joiner, 1988*; *Merino et al., 1992*; *Doorduijn et al., 2016*; *Abreu and Barbosa, 2017*), and Gram-positive bacteria are intrinsically resistant to MAC-dependent killing (*Brown, 1985*). Nevertheless, the clinical importance of the MAC in humans is made clear by recurrent infections with Gram-negative bacteria in patients treated with C5 inhibitor eculizumab (*Heesterbeek et al., 2018*) or patients with genetic deficiencies in one of the MAC components (*Lewis and Ram, 2014*).

Complement activation products that are released into plasma during complement activation are frequently used as biomarkers for infections (*Barnum et al., 2020*). One of these biomarkers is the terminal complement complex (TCC) or soluble MAC (sMAC), which is often increased in plasma of patients suffering from bacterial infections (*Lin et al., 1993*; *Mook-Kanamori et al., 2014*; *Westra et al., 2017*). However, since MAC is meant to assemble and insert in bacterial membranes, it is still unclear how sMAC is formed when complement is activated on bacteria (*Mook-Kanamori et al., 2014*). sMAC could represent debris of lysed cells, but could also represent improperly inserted MAC pores that are released from bacteria during complement activation (*Morgan et al., 2016*).

Here, we show that sMAC is primarily formed when complement is activated on bacteria that resist killing by MAC pores. A direct comparison revealed that MAC-resistant *Escherichia coli* strains generated more sMAC and converted more C5 than MAC-sensitive strains. More sMAC was also generated compared to MAC-sensitive *E. coli* when complement was activated on intrinsically MAC-resistant Gram-positive bacteria. Our data suggest that MAC did not insert into the bacterial cell envelope of MAC-resistant strains and was released from the bacterial surface. Although the release of sMAC could lyse bystander human erythrocytes in a serum-free model, serum regulators vitronectin (Vn) and clusterin (Clu) can prevent this bystander lysis. Finally, combining size exclusion chromatography (SEC) and mass spectrometry (MS) profiling of serum incubated with bacteria revealed that sMAC is a heterogeneous complex composed of C5b, C6, C7, C8, one to three copies of C9 and several copies of chaperone molecules Vn and Clu. Altogether, our study suggests that sMAC is an inactivated complex released from bacteria that resist killing by MAC.

## Results

### sMAC is primarily formed by MAC-resistant Gram-negative bacteria

To understand how sMAC is formed when bacteria activate complement, we analyzed sMAC formation by different bacteria. A panel of 12 laboratory and clinical *E. coli* strains were incubated with pooled human serum. First, we studied if these *E. coli* strains were sensitive to killing by MAC pores. Bacterial viability was assessed by counting colony forming units (CFUs) and revealed that four strains were killed in serum (*Figure 1a*). Killing was MAC-dependent because it could be inhibited with C5 inhibitors OmCI and eculizumab (*Figure 1—figure supplement 1a*), indicating that these strains are 'MAC-sensitive'. The other eight strains survived in serum (*Figure 1a*). C3 conversion was measured to determine if these serum-resistant strains activated complement at all. Deposition of C3b on the bacterial surface (*Figure 1b*) and release of C3a in the supernatant (*Figure 1—figure supplement 1b*) were similar on three strains compared to the MAC-sensitive strains, indicating that these strains activate complement efficiently. We have previously shown that MAC components bind to these strains after complement activation (*Doorduijn et al., 2021*), suggesting that they are 'MAC-resistant' strains. The other five strains showed little to no deposition of C3b and were considered 'complement-resistant'.

We next wanted to see which *E. coli* strains formed sMAC in serum using an in-house sandwich ELISA. In short, sMAC in serum was captured using an antibody recognizing a neo-epitope of C9 when it is part of sMAC (*Mollnes et al., 1985*). Next, C6 was detected with streptavidin-HRP, since we used C6-depleted serum supplemented with biotinylated C6. The specificity of our sMAC ELISA was validated using cobra venom factor (CVF) (*Figure 1—figure supplement 2a*), which can form fluid-phase

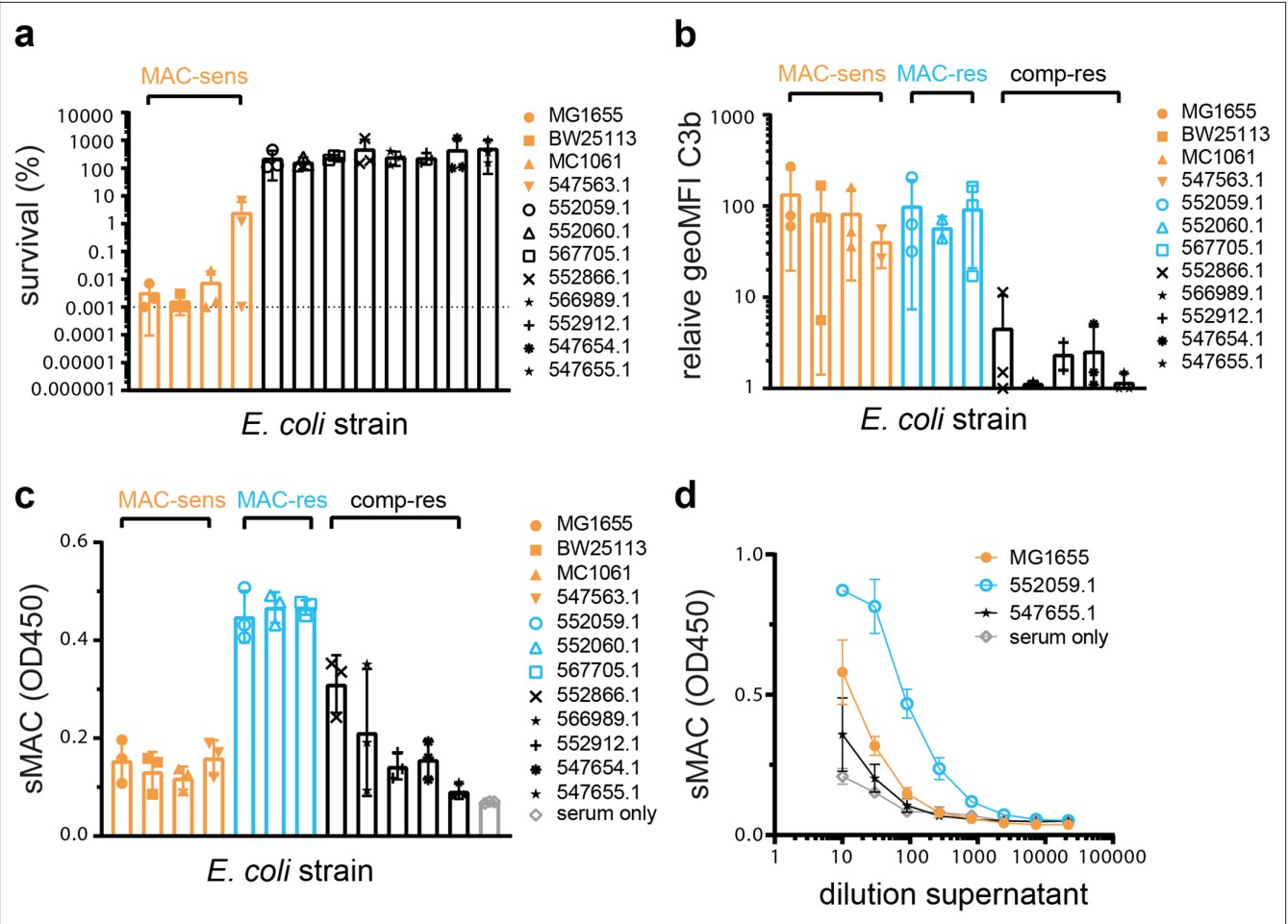

**Figure 1.** Soluble membrane attack complex (sMAC) is primarily formed by MAC-resistant Gram-negative bacteria. *Escherichia coli* strains ($5 \times 10^7$ bacteria/ml) were incubated in 5% pooled human serum. Bacterial viability was determined after 60 min by counting colony forming units (CFUs) and calculating the survival compared to t=0. The horizontal dotted line represents the detection limit of the assay. (**b**) *E. coli* strains ($5 \times 10^7$ bacteria/ml) were incubated in 10% C5-depleted serum. Bacteria were stained with AF488-labelled mouse monoclonal anti-C3b after 30 min and staining was measured by flow cytometry. The relative binding was calculated by normalizing the geoMFI to the geoMFI of unlabelled bacteria. (**c**) sMAC was detected in the reaction supernatant (100-fold diluted) by enzyme-linked immunosorbent assay (ELISA) after *E. coli* strains ($5 \times 10^8$ bacteria/ml) were incubated in 5% C6-depleted serum supplemented with C6-biotin for 60 min. Serum without bacteria (serum only) was taken as background control. (**d**) sMAC was detected by ELISA for a dilution range of reaction supernatant collected in c for *E. coli* strains MG1655, 552059.1, and 547655.1. Orange strains are MAC-sensitive (MAC-sens), blue strains MAC-resistant (MAC-res), and black strains complement-resistant (comp-res). Flow cytometry data (**b**) are represented by individual geoMFI values of the bacterial population. Data represent mean ± SD (**d**) or individual values with mean ± SD (**a, b, c**) of three independent experiments.

The online version of this article includes the following figure supplement(s) for figure 1:

**Figure supplement 1.** Complement dependency of bacterial killing and C3a release in serum for *Escherichia coli* strains.

**Figure supplement 2.** Validation specificity soluble membrane attack complex (sMAC) enzyme-linked immunosorbent assay (ELISA) and sMAC release by *Klebsiella* strains.

C5 convertases in serum (***Vogel and Fritzinger, 2010***). All tested MAC-resistant strains efficiently formed sMAC in serum, whereas MAC-sensitive strains and most complement-resistant strains only generated slightly more sMAC than present in serum alone (***Figure 1c***). Titration of the supernatant of MAC-resistant 552059.1 suggested that there was at least fivefold more sMAC compared to MAC-sensitive MG1655 or complement-resistant 547654.1 (***Figure 1d***). sMAC was formed in a complement-dependent manner, since C5 inhibitor OmCI and eculizumab prevented formation of sMAC (***Figure 1—figure supplement 2b***). Finally, two MAC-resistant *Klebsiella* strains also formed more sMAC in serum compared to MAC-sensitive *Klebsiella* strains (***Figure 1—figure supplement 2c***), suggesting that these findings can also be translated to other Gram-negative species. Altogether,

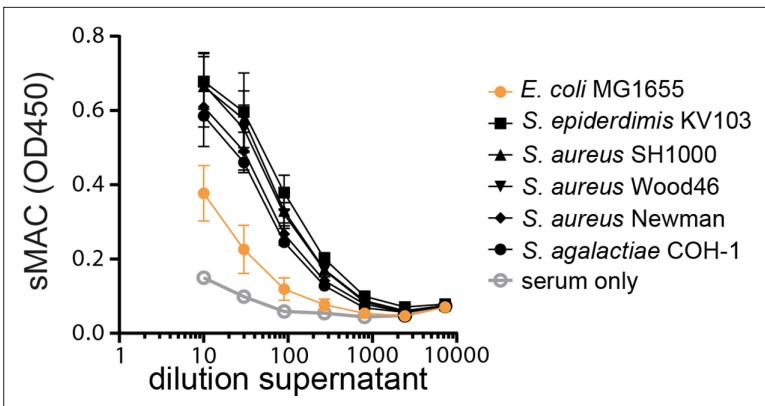

**Figure 2.** Gram-positive bacteria also form soluble membrane attack complex (sMAC) in serum. MAC-sensitive *Escherichia coli* MG1655 and Gram-positive strains ($5 \times 10^8$ bacteria/ml) were incubated in 5% C6-depleted serum supplemented with C6-biotin. The supernatant was collected after 60 min by centrifugation. Serum without bacteria (serum only) was taken as background control. sMAC was detected in a dilution range of reaction supernatant by enzyme-linked immunosorbent assay (ELISA). Data represent mean ± SD of three independent experiments.

The online version of this article includes the following figure supplement(s) for figure 2:

**Figure supplement 1.** C5a generation by Gram-positive bacteria in serum.

these data suggest that for Gram-negative bacteria sMAC is primarily formed in serum by MAC-resistant strains.

## Gram-positive bacteria also form sMAC in serum

Our data indicate that sMAC is mainly produced by MAC-resistant Gram-negative bacteria in serum. However, sMAC is also detected in plasma of patients that suffer from bacterial infections with Gram-positive bacteria (*Barnum et al., 2020*; *Lin et al., 1993*; *Mook-Kanamori et al., 2014*). Gram-positive bacteria are intrinsically resistant to MAC-dependent killing (*Brown, 1985*), which is thought to be caused by the composition of the bacterial cell envelope. Gram-negative bacteria have a cell envelope containing an outer membrane, periplasmic peptidoglycan layer, and a cytosolic IM, whereas Gram-positive bacteria only have a cytosolic membrane that is shielded by a thick peptidoglycan layer. This thick peptidoglycan layer is thought to be responsible for preventing MAC formation in the cytosolic membrane and bacterial killing (*Brown, 1985*). Here, we wanted to study if complement activation on Gram-positive bacteria also generates sMAC in serum.

Three *Staphylococcus aureus* strains (SH1000, Wood46 and Newman), one *Staphylococcus epidermidis* strain (KV103), and one *Streptococcus agalactiae* strain (COH-1) were incubated in serum to detect sMAC in the serum supernatant. We have previously shown that these strains activate complement and resist killing by MAC pores (*Berends et al., 2013*). sMAC was formed in all five Gram-positive strains (*Figure 2*). Since we could not compare this with MAC-sensitive Gram-positive bacteria because of intrinsic resistance to MAC, we compared sMAC generation with MAC-sensitive *E. coli* MG1655. sMAC generation was three- to fivefold higher for Gram-positive strains compared to MAC-sensitive *E. coli* MG1655 (*Figure 2*), corresponding more or less with the difference observed for MAC-resistant *E. coli* (*Figure 1d*). Altogether, these data indicate that Gram-positive bacteria also form sMAC in serum.

## MAC-resistant *E. coli* strains potently convert C5 in serum

Because conversion of C5 into C5b is crucial to initiate the assembly of sMAC, we wanted to know if conversion of C5 was also higher on MAC-resistant strains compared to MAC-sensitive strains in serum. To study this, we compared C5 conversion for the MAC-sensitive and MAC-resistant *E. coli* strains used in *Figure 1*. Western blotting of the serum supernatant confirmed that MAC-resistant strains converted all C5 in serum (*Figure 3a*), whereas leftover C5 was still visible for MAC-sensitive strains. A sandwich ELISA was also used to quantify the released C5a into serum supernatant, which revealed that MAC-resistant strains generated ± fivefold more C5a compared to MAC-sensitive strains

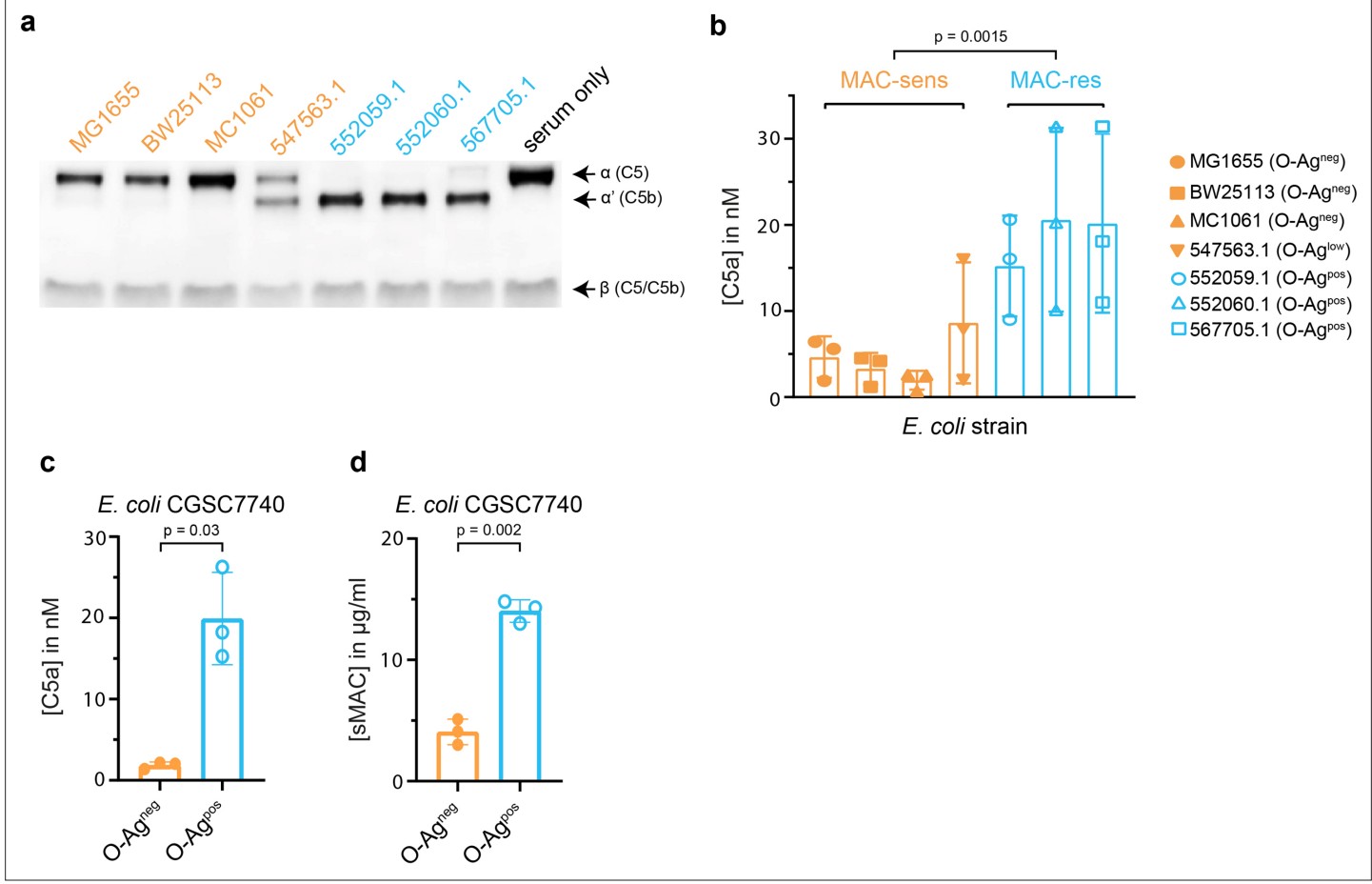

**Figure 3.** Membrane attack complex (MAC)-resistant *Escherichia coli* strains potently convert C5 in serum. *E. coli* strains ($5 \times 10^8$ bacteria/ml) were incubated in 5% pooled human serum and supernatant was collected by centrifugation after 60 min. (**a**) Representative Western blot for C5 of the supernatant. The upper band represents the α-chain of C5, the middle band of C5b (α′), and the lower band the β-chain of both C5 and C5b. Serum without bacteria (ser) was taken as control for the absence of C5 conversion. The Western blot is a representative of at least three independent experiments. (**b**) C5a in the supernatant was quantified by enzyme-linked immunosorbent assay (ELISA). Orange strains are MAC-sensitive (MAC-sens) and blue strains are MAC-resistant (MAC-res). *E. coli* CGSC7740 wildtype without lipopolysaccharide (LPS) O-antigen (O-Ag) (O-Ag$^{neg}$) and *wbbL*+ with LPS O-Ag (O-Ag$^{pos}$) were also incubated in 5% pooled human serum to collect supernatant. C5a (**c**) and sMAC (**d**) in the supernatant were quantified by ELISA. ELISA data represent individual values with mean ± SD of three independent experiments. Statistical analysis was done using an unpaired two-tailed t-test with the mean C5a concentrations of MAC-sensitive strains vs. MAC-resistant strains (**b**) or a paired two-tailed t-test on individual samples (**c and d**). Relevant p-values are indicated in the figure.

The online version of this article includes the following source data and figure supplement(s) for figure 3:

**Source data 1.** *Escherichia coli* strains ($5 \times 10^8$ bacteria/ml) were incubated in 5% pooled human serum and supernatant was collected by centrifugation after 60 min.

**Figure supplement 1.** Lipopolysaccharide (LPS) O-antigen (O-Ag) expression of Gram-negative strains and its effect on C3b deposition.

**Figure supplement 1—source data 1.** *Escherichia coli* strains were typed for the presence of lipopolysaccharide (LPS) O-antigen (O-Ag) via silver staining (methods described in *Doorduijn et al., 2021*).

**Figure supplement 1—source data 2.** *Klebsiella* strains were typed for the presence of lipopolysaccharide (LPS) O-antigen (O-Ag) via silver staining (methods described in *Doorduijn et al., 2021*).

(*Figure 3b*). This was comparable to the difference in sMAC (*Figure 1d*). Gram-positive bacteria that generated sMAC (*Figure 2*) also generated more C5a compared to MAC-sensitive MG1655 (*Figure 2—figure supplement 1*), although this difference in C5a generation was smaller (±threefold) compared to the difference between MAC-resistant and MAC-sensitive *E. coli* (±tenfold, *Figure 3b*). One MAC-sensitive strain (547563.1) converted more C5 and released more C5b into the supernatant compared to the other MAC-sensitive strains (*Figure 3a and b*), although this difference was not

detected by sMAC ELISA (*Figure 1c*). Nonetheless, these data suggest that MAC-resistant bacteria potently convert C5 in serum.

## Expression of LPS O-Ag on *E. coli* increases C5a and sMAC generation in serum

We wanted to further study how MAC-resistance in *E. coli* could affect C5 conversion and sMAC generation. We wondered if differences in the expression of lipopolysaccharide (LPS) O-antigen (O-Ag) could contribute. LPS O-Ag is an important constituent of the outer membrane of Gram-negative bacteria that has frequently been associated with MAC resistance (*Grossman et al., 1987*). We have previously shown that the three tested MAC-resistant strains express O-Ag (*Doorduijn et al., 2021*), whereas only one out of four MAC-sensitive strains (547563.1) does as well (shown in *Figure 3—figure supplement 1a*, summarized in Table 1). Silver staining of O-Ag for *Klebsiella* strains suggested a comparable trend, showing little detectable O-Ag for MAC-sensitive strains compared to MAC-resistant strains (*Figure 3—figure supplement 1b*). To more directly study if LPS O-Ag affects C5a generation and sMAC release, a MAC-sensitive *E. coli* K12 strain without O-Ag (O-Ag$^{neg}$) was incubated in 5% human serum and compared with an isogenic MAC-resistant strain in which O-Ag expression is restored (O-Ag$^{pos}$) (*Doorduijn et al., 2021*). Expression of O-Ag increased C5a generation in the supernatant 10-fold (*Figure 3c*) and sMAC release 3.5-fold (*Figure 3d*). C3b deposition on the bacterial surface was comparable both in the presence and absence of O-Ag in serum (*Figure 3—figure supplement 1c*), suggesting that initial complement activation was not affected by the expression of O-Ag, similar to other MAC-resistant *E. coli* (*Figure 1b*). Therefore, these data indicate that expression of O-Ag on *E. coli* can increase C5a generation and sMAC release in serum.

## Binding of C8 and C9 triggers release of MAC precursors from MAC-resistant *E. coli*

We next studied at what stage of MAC assembly the nascent MAC is released from the bacterial surface. MAC pores assemble in a stepwise manner. C5b binds to C6 to form a stable C5b6 complex, which next binds C7 to anchor the C5b-7 complex to the membrane of bacteria (*Preissner et al., 1985*). Finally, binding of C8 inserts the nascent MAC into the bacterial cell envelope, and is more tightly inserted when C9 binds and polymerizes a transmembrane ring (*Bayly-Jones et al., 2017*). Release of C5b into the supernatant was therefore measured in the presence or absence of downstream MAC components for both MAC-resistant *E. coli* 552059.1 and MAC-sensitive *E. coli* MG1655. Bacteria were labelled with convertases in C5-depleted serum and washed as done previously (*Heesterbeek et al., 2019*). Next, C5 and C6 were added in the presence or absence of downstream MAC components (*Figure 4a*). Western blotting of the supernatant revealed that more C5 was converted into C5b for MAC-resistant 552059.1 compared to MAC-sensitive MG1655 (*Figure 4b*, indicated by the orange and blue arrow), in line with *Figure 4b*. Western blotting revealed that binding of C7 to the nascent MAC prevented release of C5b6 from the bacterial surface (*Figure 4b*), as was previously observed for MAC-sensitive MG1655 (*Doorduijn et al., 2020*). Binding of C7 also appeared to increase C5 conversion on MAC-resistant 552059.1 (*Figure 4b*). However, binding of C8 to the nascent MAC triggered release of C5b from MAC-resistant 552059.1, even in the presence of final MAC component C9 (*Figure 4b*). Binding of C8 triggered some release of C5b from the surface of MAC-sensitive MG1655, but this was prevented when C9 was also present (*Figure 4b*). Quantification of C5b6 by ELISA revealed that binding of C8 and C9 released fourfold more C5b6 from the surface of MAC-resistant *E. coli* (*Figure 4c and d*). These data suggest that binding of C8 and C9 to C5b-7 triggers release of the nascent MAC from MAC-resistant *E. coli*.

## Release of MAC precursors from *E. coli* triggers lysis of bystander human erythrocytes, but is prevented by serum regulators Vn and Clu

Although we measured the release of C5b (*Figure 4*), the composition of the released complexes and their capacity to lyse cells remained unclear. Previous reports showed that complement activation on erythrocytes can cause MAC-dependent lysis of bystander cells that are not recognized by the complement system, the so-called bystander lysis (*Lachmann and Thompson, 1970*; *Götze and Müller-Eberhard, 1970*; *Cooper and Müller-Eberhard, 1970*). We next tested whether complement activation and subsequent release of MAC precursors from MAC-resistant *E. coli* can also cause

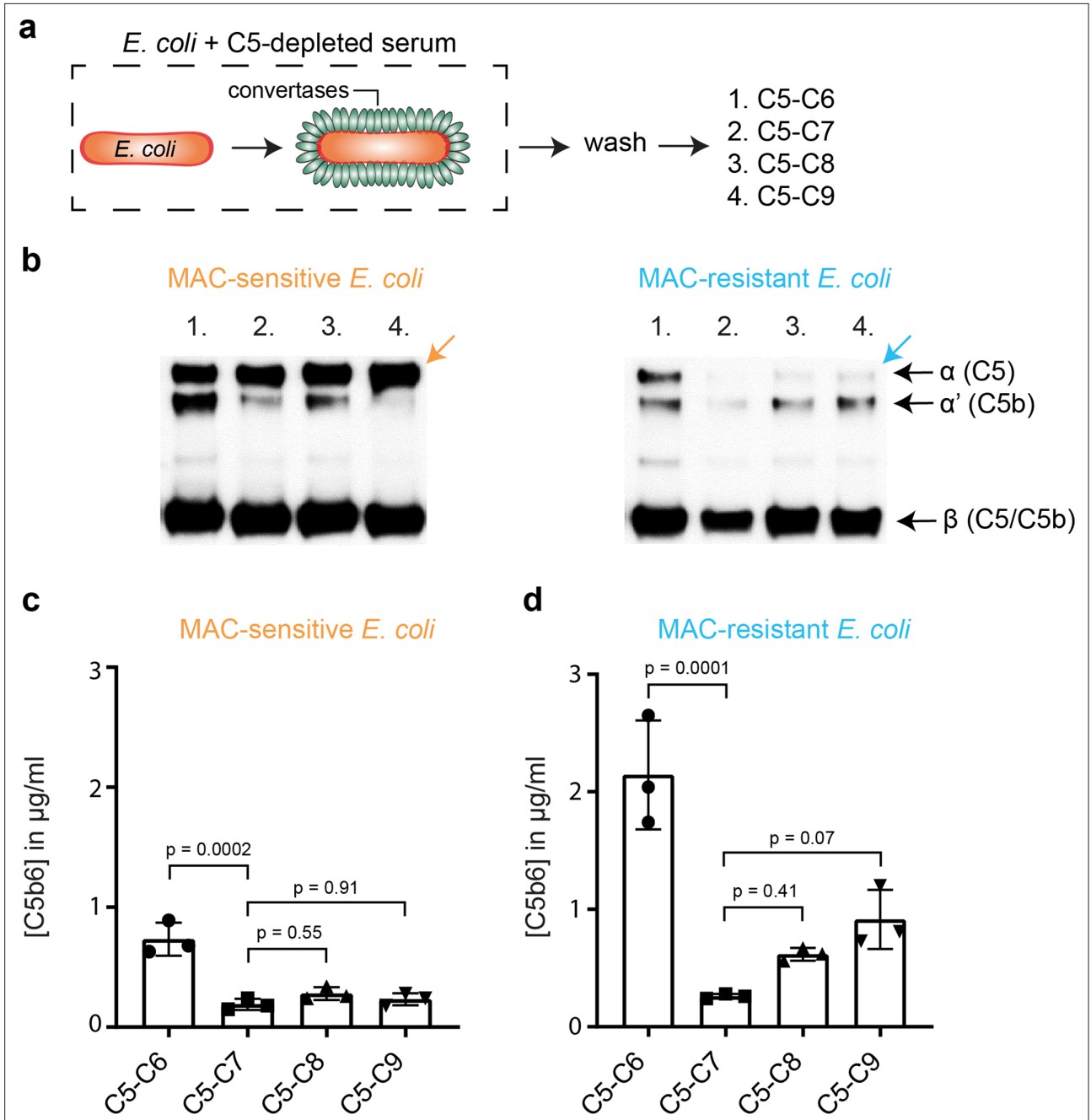

**Figure 4.** Binding of C8 and C9 triggers release of membrane attack complex (MAC) precursors from MAC-resistant *Escherichia coli*. (**a**) Schematic overview of how *E. coli* (orange rods) MG1655 (MAC-sensitive) and 552059.1 (MAC-resistant) were labelled with convertases (green ovals) in 10% C5-depleted serum. Next, bacteria were washed and bacteria ($5\times10^8$ bacteria/ml for b, $1\times10^8$ bacteria/ml for c and d) were incubated with alternative pathway (AP) convertase components (5 µg/ml FB and 0.5 µg/ml FD) and 100 nM C5 and C6 (1); 100 nM C5, C6, and C7 (2); 100 nM C5, C6, C7, and C8 (3) or 100 nM C5, C6, C7, C8, and 1000 nM C9 (4). The supernatant was collected after 60 min by centrifugation. (**b**) Western blot for C5 of the supernatant. The Western blot is a representative of at least three independent experiments. The upper band represents the α-chain of C5, the middle band of C5b (α'), and the lower band the β-chain of both C5 and C5b. C5b6 in the supernatant of MAC-sensitive MG1655 (**c**) and MAC-resistant 552059.1 (**d**) was quantified by enzyme-linked immunosorbent assay (ELISA). Dotted line represents the background OD450. ELISA data represent individual values with mean ± SD of three independent experiments. Statistical analysis was done using an ordinary one-way ANOVA with Tukey's multiple comparisons test (**c and d**) and relevant p-values are indicated in the figure.

The online version of this article includes the following source data for figure 4:

**Source data 1.** *Escherichia coli* MG1655 and 552059.1 were labelled with convertases in 10% C5-depleted serum.

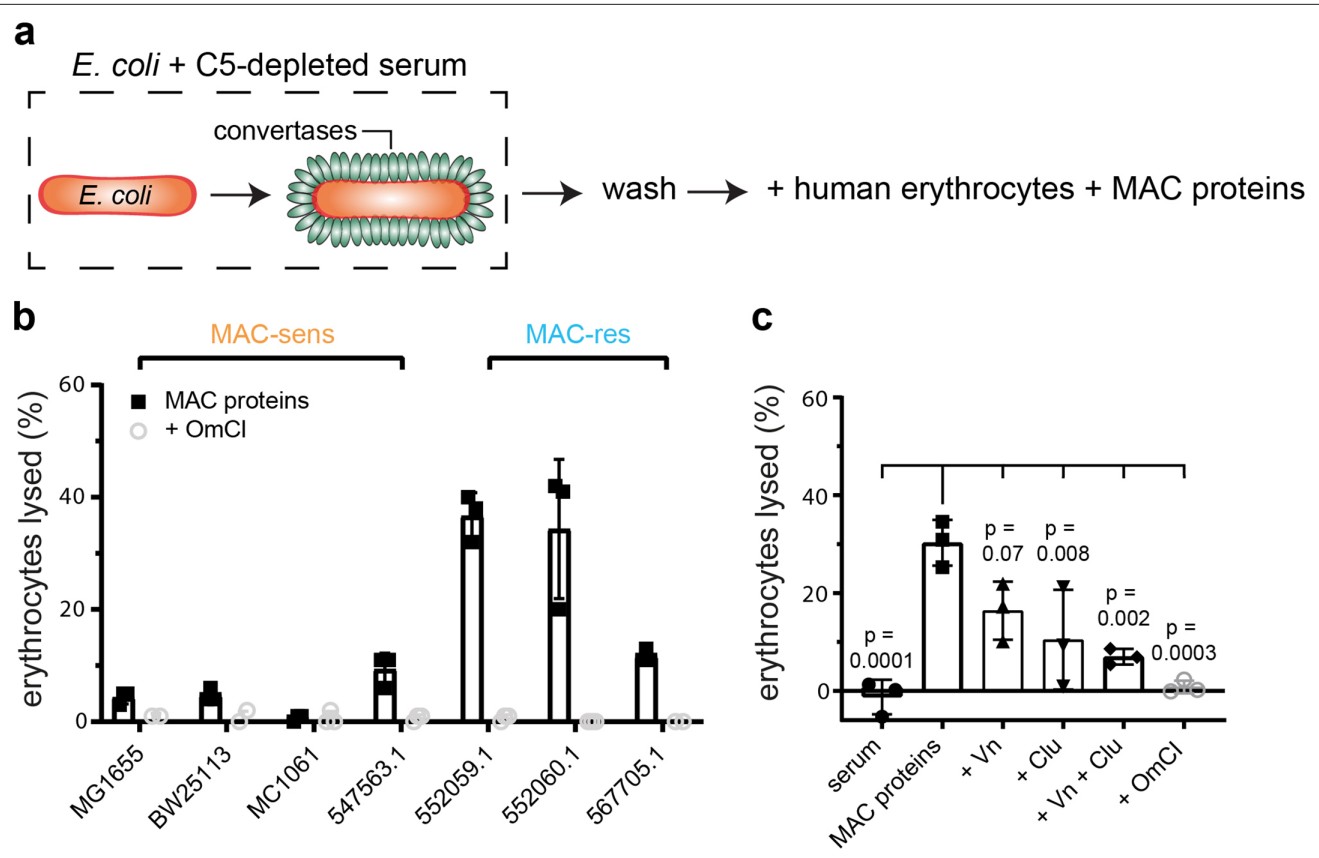

**Figure 5.** Release of membrane attack complex (MAC) precursors from *Escherichia coli* triggers lysis of bystander human erythrocytes, but is prevented by serum regulators vitronectin (Vn) and clusterin (Clu). (**a**) Schematic overview of the bystander lysis assay. *E. coli* strains (orange rods) were labelled with convertases (green ovals) in 10% C5-depleted serum and washed. Next, convertase-labelled bacteria ($3.3 \times 10^8$ per ml) were incubated with: human erythrocytes ($1 \times 10^8$ per ml), alternative pathway (AP) convertase components (5 nM FB and 20 nM FD) and MAC proteins (100 nM C5, 100 nM C6, 100 nM C7, 100 nM C8, and 500 nM C9). The supernatant was collected after 60 min by centrifugation and analyzed for the presence of hemoglobulin. The percentage of lysed erythrocytes was calculated by setting a buffer-only control at 0% lysis and MilliQ control at 100% lysis. (**b**) Bystander erythrocyte lysis for MAC-sensitive (MAC-sens) and MAC-resistant (MAC-res) *E. coli* strains. (**c**) Bystander erythrocyte lysis for convertase-labelled MAC-resistant *E. coli* 552059.1 incubated with 10% pooled human serum, MAC proteins (30 nM C5, 30 nM C6, 30 nM C7, 30 nM C8, and 300 nM C9) or MAC components with 133 nM Vn, 133 nM Clu, or 20 μg/ml C5 conversion inhibitor OmCI. Data represent individual values with mean ± SD of three independent experiments. Statistical analysis was done using an ordinary one-way ANOVA with Tukey's multiple comparisons test (**c**) and relevant p-values are indicated in the figure (all conditions compared with MAC proteins only).

The online version of this article includes the following source data and figure supplement(s) for figure 5:

**Figure supplement 1.** Soluble membrane attack complex (sMAC) release with purified MAC proteins from convertase-labelled *Escherichia coli* strains.

**Figure supplement 2.** Effect of vitronectin (Vn) and clusterin (Clu) on C9 polymerization and target-specific membrane attack complex (MAC) assembly.

**Figure supplement 2—source data 1.** Membrane attack complex (MAC)-resistant *Escherichia coli* 552059.1 was labelled with convertases in 10% C5-depleted serum and washed.

bystander lysis. Therefore, *E. coli* were labelled with convertases in C5-depleted serum as described in **Figure 4a**. Next, these convertase-labelled bacteria were incubated with purified MAC components and unlabelled human erythrocytes to measure bystander lysis (**Figure 5a**). This resulted in lysis of bystander erythrocytes for all MAC-resistant *E. coli* strains and MAC-sensitive 547563.1 (**Figure 5b**), corresponding with the production of sMAC (**Figure 5—figure supplement 1**). Lysis was prevented in the presence of C5 conversion inhibitor OmCI (**Figure 5b**), suggesting that lysis was MAC-dependent. These data show that release of MAC precursors from *E. coli* can result in lysis of bystander human cells. However, when we studied bystander lysis in a human serum environment, we observed that the 552059.1 did not trigger bystander lysis of erythrocytes (**Figure 5c**). Serum regulators Vn and Clu are known to scavenge and inactivate sMAC (**Zipfel and Skerka, 2009**; **Schmidt et al., 2016**). Indeed, both Vn and Clu inhibited bystander lysis of erythrocytes when MAC assembled

on convertase-labelled *E. coli* 552059.1 (as described in *Figure 5a*) at concentrations representative for 10% serum (*Figure 5c*). SDS-PAGE revealed that Clu, but not Vn, prevents the formation of polymeric-C9 in the supernatant (*Figure 5—figure supplement 2a*). This suggests that Vn and Clu interfere at different stages in the assembly of sMAC. Vn and Clu both specifically prevent lysis of bystander cells by MAC, since Vn and Clu did not inhibit binding of C9 (*Figure 5—figure supplement 2b*), or MAC-dependent killing (*Figure 5—figure supplement 2c*) when MAC was assembled by local conversion of C5 on convertase-labelled MAC-sensitive MG1655. Altogether, our data suggest that release of MAC precursors from *E. coli* can trigger lysis of bystander human erythrocytes, but that serum regulators Vn and Clu can inhibit this bystander lysis.

## sMAC that is released from bacteria is a heterogeneous protein complex with different stoichiometries

Next, we aimed to define the molecular composition of sMAC generated when complement is activated on bacteria. sMAC was generated by incubating MAC-resistant *E. coli* 552059.1 in C6-depleted serum with His-tagged C6 and captured and isolated with HisTrap beads (*Figure 6—figure supplement 1a*). SEC was used to separate sMAC from monomeric-C6. The SEC profile of serum incubated with MAC-resistant *E. coli* shifted to much shorter elution times compared to nonactivated serum (*Figure 6a*, fractions B5-B12), indicating a mass shift. As a control, we analyzed commercially available sMAC, which is generated with zymosan particles in serum. Commercial sMAC eluted from the SEC column in the same fractions (B5-B12), indicating that these bacterial-eluate fractions contain sMAC. Blue-native PAGE (BN-PAGE) and subsequent Western blotting for C6 and C9 confirmed that these fractions contain sMAC (*Figure 6b*). Compared to commercial sMAC, bacterial sMAC eluted somewhat later from the SEC column (*Figure 6b*, most apparent in fractions B11 and B12). In addition, sMAC complexes generated by *E. coli* seemed to run further into the gel. These data suggest that bacterial sMAC complexes have a different composition compared to commercial sMAC.

*Menny et al., 2021* recently reported that the commercially available sMAC used in our study consists of C5b-8, one to three copies of C9 and several copies of Vn and Clu, using proteomics, cross-linking MS, and cryo-electron microscopy. Here, we compared the average composition of sMAC generated by MAC-resistant *E. coli* with commercial sMAC by profiling sMAC components with liquid chromatography-tandem mass spectrometry (LC-MS/MS). The total amount of sMAC components that were detected with LC-MS/MS in individual fractions (*Figure 6c* and *Figure 6—figure supplement 1b,c*) corresponded with BN-PAGE (*Figure 6b*), suggesting that most bacterial sMAC eluted later from the SEC column than commercial sMAC. For both bacterial and commercial sMAC, sMAC components C5, C7, and C8 were present in equal amounts in a pooled sample of fractions B5-B12 (*Figure 6d*). However, C6 and Vn were both twofold more abundant for bacterial sMAC (*Figure 6d*). The increased ratio of C6 is likely explained by the fact that not all monomeric-C6 could be separated during SEC (as was visible in fraction B9-B12 by BN-PAGE in *Figure 6b*). Surprisingly, bacterial sMAC contained on average less C9 per C5 (*Figure 6d*, ratio of 2:1) than commercially available sMAC (*Figure 6d*, ratio of 3:1). The relative amount of C9 per C5 (*Figure 6e*), but not other sMAC components (*Figure 6—figure supplement 1d*), decreased in fractions that eluted later from the SEC column for bacterial sMAC. This was not observed for commercially available sMAC (*Figure 6f* and *Figure 6—figure supplement 1e*). These data suggest that sMAC generated by bacteria contains more complexes with less C9 compared to commercial sMAC.

Finally, we assessed the relative abundance of sMAC components for sMAC that was generated by Gram-positive bacteria. *S. aureus* Wood46 was incubated with human serum and the supernatant was separated by SEC (without isolating sMAC via HisTrap beads), using nonactivated serum as control. By profiling sMAC components in individual fractions by LC-MS/MS, we found that in nonactivated serum these components elute later in the SEC profile, corresponding to monomeric or low molecular weight complexes (*Figure 7*). Upon activation, the profiles of all sMAC components largely co-elute and are shifted to fractions that correspond to the higher mass range. In fact, close to no monomeric MAC components were detected after incubation with bacteria, suggesting that all available MAC components were incorporated into hetero-oligomeric sMAC complexes. We did not observe any sMAC without Vn and Clu in the bacterial activated sample, since these would be located around the 664 kDa molecular weight marker, suggesting that all sMAC complexes were bound to multiple Vn and Clu molecules (corresponding with the relative abundance in *Figure 6d*). The complexes correlate

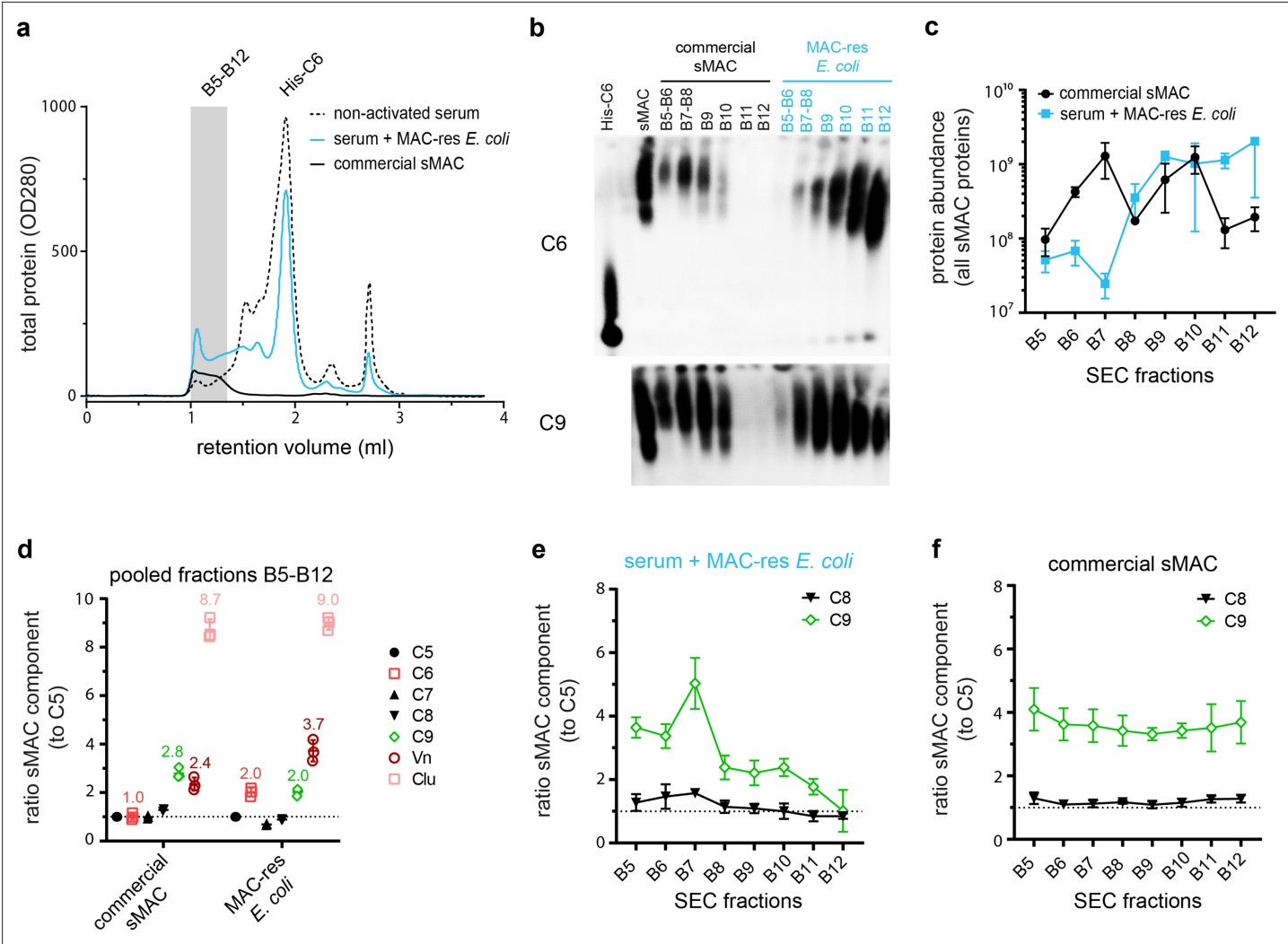

**Figure 6.** Isolation of soluble membrane attack complex (sMAC) using His-tagged C6 and analysis by blue-native PAGE (BN-PAGE) and liquid chromatography-tandem mass spectrometry (LC-MS/MS). sMAC was generated by incubating MAC-resistant (MAC-res) *Escherichia coli* 552059.1 in C6-depleted serum supplemented with His-tagged C6 (His-C6). sMAC in the supernatant was captured with HisTrap beads and eluted. (**a**) Concentrated eluate was separated by size exclusion chromatography (SEC) on a Superose 6 column and OD280 was measured to determine protein content (OD280). Eluate from serum without bacteria (nonactivated serum) and 50 µg commercially available sMAC (Complement Technology) were analyzed as controls. (**b**) BN-PAGE was performed with pooled (B5+B6 and B7+B8) or individual (B9-B12) SEC fractions and analyzed by Western blotting for C6 (above) and C9 (below). Black fractions represent SEC fractions commercially available sMAC, blue fractions represent SEC fractions of serum incubated with MAC-res *E. coli*. Two µg of purified sMAC and His-C6 were loaded as control. (**c**) The protein abundance of all sMAC components (iBAQ value) was determined by LC-MS/MS for individual SEC fractions (B5–B12). (**d**) The ratio of all individual sMAC components to C5 was determined in a pooled sample of fraction B5-B12. The mean ratio of individual samples was indicated for relevant components above the dot plots. (**e**) In addition, the ratio of C8 and C9 to C5 was determined for each individual fraction separately for serum incubated with MAC-res *E. coli* and commercially available sMAC (**f**). The dotted line (**d, e, and f**) represents a ratio of 1. The SEC profile and Western blot are representative of three independent experiments. LC-MS/MS data represent three individual digests of the same fraction with mean ± SD that are representative for two independent experiments.

The online version of this article includes the following source data and figure supplement(s) for figure 6:

**Source data 1.** Soluble membrane attack complex (sMAC) was generated by incubating MAC-resistant (MAC-res) *Escherichia coli* 552059.1 in C6-depleted serum supplemented with His-tagged C6 (His-C6).

**Figure supplement 1.** Validation isolation soluble membrane attack complex (sMAC) with HisTrap beads and liquid chromatography-tandem mass spectrometry (LC-MS/MS) size exclusion chromatography (SEC) fractions.

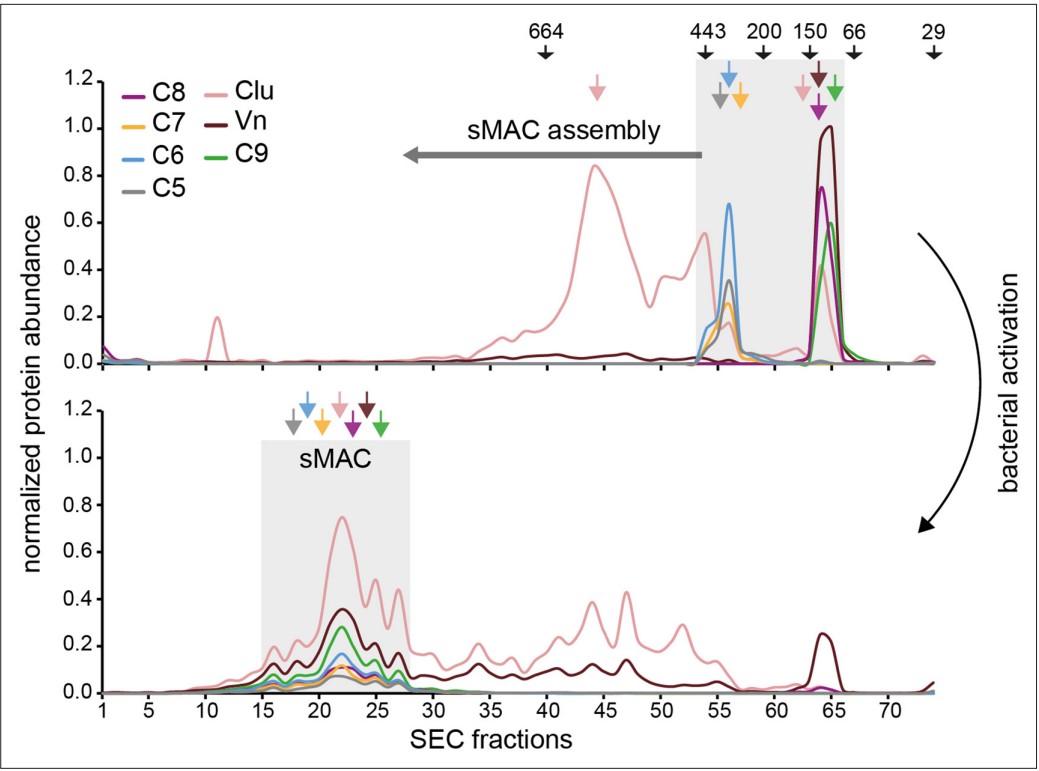

**Figure 7.** Soluble membrane attack complex (sMAC) that is released from bacteria is a heterogeneous protein complex with different stoichiometries. Mass spectrometry (MS) profiling of sMAC components in serum supernatant incubated for 3 hr at 37°C with (bottom, activated) and without (top, nonactivated) *Staphylococcus aureus* Wood46. Serum was separated by size exclusion chromatography (SEC) and the protein abundance (normalized iBAQ values) in each fraction was determined by liquid chromatography-tandem mass spectrometry (LC-MS/MS). The gray boxes indicate the elution of sMAC components in nonactivated and activated serum. The arrows on the top indicates elution of molecular weight (kDa) markers.

with a ratio of one to three C9 molecules and an average of two C9 molecules to C5b-8, with Clu as the most abundant regulator bound to sMAC. Altogether, these data confirm that sMAC released from bacteria is a heterogeneous assembly comprising a single copy of C5b-8 together with multiple copies of C9, Clu and Vn in a mixture of stoichiometries.

## Discussion

Although plasma levels of sMAC are frequently increased during bacterial infections (***Barnum et al., 2020***; ***Lin et al., 1993***), it remains largely unknown how sMAC is formed during infections and what the complex represents. Here, we show that sMAC is an inactivated complex that is released from bacteria during complement activation. We show that sMAC is primarily released from MAC-resistant bacteria, including Gram-positive bacteria (***Figures 1 and 2***). Surprisingly, these MAC-resistant bacteria also potently activate the complement cascade and convert more C5 than MAC-sensitive bacteria.

These findings suggest that detection of sMAC in human serum indicates potent C5 conversion by MAC-resistant bacteria. Increased C5 conversion has previously been associated with MAC resistance on Gram-negative bacteria (***Joiner et al., 1982a***; ***Krukonis and Thomson, 2020***), but this has not yet been directly linked to the detection of sMAC in human serum. Why C5 conversion is increased on bacteria that resist killing by MAC pores remains unclear. On Gram-negative bacteria, we observed that potent C5 conversion by MAC-resistant *E. coli* is linked to the expression LPS O-Ag (***Figure 3***). As Gram-positive bacteria do not express LPS O-Ag and are inherently MAC-resistant, a similar comparison could not be made for Gram-positive bacteria. Nonetheless, this suggests that cell envelope constituents, and in the case of Gram-negative bacteria specifically LPS O-Ag, can affect total C5

conversion. Grossman et al. have previously demonstrated that linking *Salmonella* LPS O-Ag to sheep erythrocyte membranes can increase C3 consumption (*Grossman et al., 1990*). However, in our study, no difference was detected in C3 conversion and C3b deposition, suggesting that initial recognition and complement activation was comparable for MAC-sensitive and MAC-resistant bacteria. For the conversion of C5, however, the density of C3b is known to be important (*Rawal and Pangburn, 2001*; *Berends et al., 2015a*). Although C3b deposition was comparable for MAC-sensitive and MAC-resistant *E. coli*, we cannot exclude that local densities of C3b might differ, which could affect C5 convertase activity and explain a difference in C5 conversion.

Our study provides molecular insight into how sMAC is formed when complement is activated on bacteria. Although it was generally believed that sMAC is formed on the bacterial surface and then released (*Morgan et al., 2016*), it was still unclear if the complete complex is formed on the bacterial surface. Our findings suggest that sMAC is initially formed on the bacterial surface, but released as the nascent MAC further assembles. Especially binding of C8 and C9 to C5b-7 triggered release of the nascent MAC from the bacterial surface (*Figure 4*). Our MS data show that most sMAC complexes ultimately contain on average one to three C9 molecules and a varying amount of Vn and Clu copies, which implies that sMAC further assembles in solution. Joiner et al. have previously observed that binding of C8 to C5b-7 on a MAC-resistant *Salmonella minnesota* strain triggered release of C5b-8 (*Joiner et al., 1982b*). Although binding of C8 to soluble C5b-7 prevents binding to membranes (*Nemerow et al., 1979*), our data suggest that released MAC precursors can still bind to bystander cells and form lytic MAC pores. It is possible that C5b-8 can immediately bind bystander cells in our assays, which prevents C5b-7 from entering a soluble state. However, we cannot exclude that trace amounts of intermediate MAC precursors, such as C5b6, that are also released from the bacterial surface and can still form enough MAC pores to lyse bystander erythrocytes.

Since C8 and C9 are the MAC components that insert into membranes (*Menny et al., 2018*; *Sharp et al., 2016*), our findings suggest that the nascent MAC is released because it is less capable of inserting into the bacterial cell envelope of MAC-resistant bacteria. Why MAC insertion is impaired on MAC-resistant bacteria remains unclear. On Gram-positive bacteria, it is believed that the dense peptidoglycan layer prevents insertion of MAC into the cytoplasmic membrane (*Brown, 1985*). On Gram-negative bacteria, MAC resistance and improper insertion of MAC into the bacterial outer membrane have previously been linked to the expression and length of the LPS O-Ag (*Grossman et al., 1987*). We here show that the expression of LPS O-Ag directly increases the generation and release of sMAC (*Figure 3*). LPS O-Ag could prevent MAC proteins from inserting into hydrophobic patches of the bacterial outer membrane, ultimately resulting in release of sMAC. This is in line with our previous study, which demonstrated that expression of O-Ag impairs polymerization of C9 at the bacterial surface (*Doorduijn et al., 2021*). The presence of LPS O-Ag could also prevent binding of complement-activating antibodies that bind to epitopes close to the bacterial surface (*Russo et al., 2009*). Instead, antibodies that recognize the O-Ag could activate complement further away from the surface, which prevents insertion of MAC proteins into outer membrane and results in release of sMAC.

Findings in this study also highlight the importance of Vn and Clu in preventing lysis of bystander host cells when bacteria activate complement. In the absence of Vn and Clu, release of MAC precursors from bacteria resulted in lysis of bystander human erythrocytes (*Figure 5*). Importantly, Vn and Clu did not prevent MAC-dependent killing of the target bacterium, suggesting that Vn and Clu specifically inhibit bystander lysis. This bystander lysis has been studied using sensitized and unsensitized erythrocytes in the past (*Lachmann and Thompson, 1970*; *Götze and Müller-Eberhard, 1970*; *Cooper and Müller-Eberhard, 1970*), but relatively few reports have studied bystander lysis in a bacterial context (*Geelen et al., 1992*; *Verduin et al., 1994*). In these reports, complement activation on *Streptococcus pneumoniae* (*Geelen et al., 1992*) and *Moraxella catarrhalis* (*Verduin et al., 1994*) caused bystander lysis of chicken erythrocytes. We here extend on these findings showing that human erythrocytes are also sensitive to bystander lysis.

This raises the question if bystander lysis is a clinically relevant process during bacterial infections. Under physiological conditions, Vn and Clu are present in a two- to eightfold molar excess compared to MAC components in serum, favoring inactivation of released MAC over binding to a bystander cell membrane (*Barnum et al., 2020*). Correspondingly, we do not observe bystander lysis in a serum of healthy donors in our study. However, *Willems et al., 2019* have recently reported that Clu is

decreased in plasma of children suffering from bacterial infections. Vn and Clu polymorphisms have also been identified that impair their potency in scavenging and inactivating sMAC precursors (*Ståhl et al., 2009*; *van den Heuvel et al., 2018*), which could predispose to host cell damage by bystander lysis. For both polymorphisms, patients suffered from recurrent complement-mediated hemolytic uremic syndrome, and in case of the Vn polymorphism, this was even associated with recurrent *E. coli* infections. Moreover, both Gram-negative and Gram-positive bacterial pathogens are known to recruit Vn to their surface (*Singh et al., 2010*; *Riesbeck, 2020*). For Gram-negative bacteria, this is thought to prevent MAC-dependent killing (*Hallström et al., 2009*; *Griffiths et al., 2011*), but for Gram-positive bacteria, the purpose of recruiting Vn and Clu has remained elusive. Altogether, our study therefore suggests a potential role of bystander lysis during bacterial infections that merits further investigation.

Finally, we demonstrate that sMAC generated by bacteria is a heterogeneous mixture of alike protein complexes with different stoichiometries. sMAC is believed to consist of C5b-7, C8, multiple copies of C9, and several copies of Vn and Clu (*Barnum et al., 2020*; *Preissner et al., 1989*). This was recently supported by *Menny et al., 2021*, who revealed that sMAC generated by zymosan particles in serum contains at least one to three copies of C9. Our data suggest that sMAC generated by bacteria is present in similar stoichiometries, but on average, contains less C9 molecules, especially for sMAC generated by MAC-resistant *E. coli* (*Figures 6 and 7*). This suggests that the stoichiometry of sMAC could depend on the target cell that activates complement. However, it is important to note the possibility that complexes with less C9 are lost during purification of commercially available sMAC. Our data also show the presence of multiple copies of Vn and Clu in sMAC generated by bacteria, with Clu being the most abundant chaperone. These findings suggest that Vn and Clu can efficiently capture all sMAC that is released in serum. This is in line with the recent structure of sMAC by *Menny et al., 2021*, which revealed that Clu binds and traps the terminal C9 in an intermediate conformation, thereby preventing further C9 polymerization. This is also in line with our results showing that Clu, and not Vn, is able to inhibit C9 polymerization. Finally, compared to commercial sMAC, more Vn was detected in sMAC generated by bacteria. It is unclear if this is caused by a difference in the target cell that activates complement, or a difference in the serum that was used to generate sMAC. The concentration of Vn in serum can vary largely between individuals (*Barnum et al., 2020*) and could be responsible for the observed difference.

Altogether, we show that sMAC represents an inactivated complex that is primarily released from MAC-resistant bacteria that potently activate complement. These findings are clinically relevant as they provide insight into what sMAC as a biomarker could represent during bacterial infections. Future clinical studies could determine if sMAC detection in the plasma of patients that suffer from bacterial infections also correlates with potent complement activation and MAC resistance of the causative bacterial pathogen. These insights could enhance our understanding of the role of complement activation in the pathogenesis of bacterial infections.

## Materials and methods

### Serum and complement proteins

Pooled human serum was obtained from healthy volunteers as previously described (*Berends et al., 2015b*). Serum depleted of complement components C5 or C6 was obtained from Complement Technology. CVF was obtained from Quidel. Preassembled C5b6, C8, and sMAC (SC5b-9) were obtained from Complement Technology. His-tagged C5, C6, C7, and factor B (FB) were expressed in HEK293E cells at U-Protein Express as described previously (*Doorduijn et al., 2020*). Factor D (FD) and OmCI were produced in HEK293E cells at U-Protein Express and purified as described before (*Nunn et al., 2005*). To produce fluorescently labelled C9, C9-3xGGGGS-LPeTG-6xHis was recombinantly expressed in Expi293F cells and site-specifically labelled with Cy5 via C-terminal sortagging as done previously (*Heesterbeek et al., 2019*). Biotinylated C6 was produced in a similar manner, by expressing and isolating C6-LPeTGG-6xHis (previously described in *Doorduijn et al., 2020*) and subsequent C-terminal sortagging with GGGK-biotin (kindly provided by Louris Feitsma, Department of Crystal and Structural Chemistry, Bijvoet Institute). Eculizumab was kindly provided by Genmab. Vn (plasma isolated) was obtained from Advanced Biomatrix and recombinantly expressed human Clu from R&D Systems. Monoclonal mouse-anti C3b (bH6, kindly provided by Peter Garred) was randomly

**Table 1.** Bacterial strains used in this study.

| Strain | Origin | Amount of detectable LPS O-Ag |
|---|---|---|
| *Escherichia coli* MG1655 | Laboratory strain | Absent (*Doorduijn et al., 2021*) (also in *Figure 3—figure supplement 1*) |
| *Escherichia coli* BW25113 | Laboratory strain | Absent (*Doorduijn et al., 2021*) (also in *Figure 3—figure supplement 1*) |
| *Escherichia coli* MC1061 | Laboratory strain | Absent (*Doorduijn et al., 2021*) (also in *Figure 3—figure supplement 1*) |
| *Escherichia coli* 547563.1 | Clinical isolate* | Low[†] (*Doorduijn et al., 2021*) (also in *Figure 3—figure supplement 1*) |
| *Escherichia coli* 552059.1 | Clinical isolate* | High (*Doorduijn et al., 2021*) (also in *Figure 3—figure supplement 1*) |
| *Escherichia coli* 552060.1 | Clinical isolate* | High (*Doorduijn et al., 2021*) (also in *Figure 3—figure supplement 1*) |
| *Escherichia coli* 567705.1 | Clinical isolate* | High (*Doorduijn et al., 2021*) (also in *Figure 3—figure supplement 1*) |
| *Escherichia coli* 566989.1 | Clinical isolate* | High (*Figure 3—figure supplement 1*) |
| *Escherichia coli* 552912.1 | Clinical isolate* | No (*Figure 3—figure supplement 1*) |
| *Escherichia coli* 552866.1 | Clinical isolate* | High (*Figure 3—figure supplement 1*) |
| *Escherichia coli* 547654.1 | Clinical isolate* | High (*Figure 3—figure supplement 1*) |
| *Escherichia coli* 547655.1 | Clinical isolate* | High (*Figure 3—figure supplement 1*) |
| *Klebsiella variicola* 402 | Clinical isolate* | Low[†] (*Figure 3—figure supplement 1*) |
| *Klebsiella pneumoniae* 567880.1 | Clinical isolate* | Low[†] (*Figure 3—figure supplement 1*) |
| *Klebsiella pneumoniae* 567702.1 | Clinical isolate* | High (*Figure 3—figure supplement 1*) |
| *Klebsiella pneumoniae* 567709.1 | Clinical isolate* | High (*Figure 3—figure supplement 1*) |
| *Escherichia coli* CGSC7740 | Laboratory strain | Absent (*Doorduijn et al., 2021*) |
| *Escherichia coli* CGSC7740 *wbbL*+ | Laboratory strain | High (*Doorduijn et al., 2021*) |
| *Staphylococcus aureus* SH1000 | Laboratory strain | n.a. |
| *Staphylococcus aureus* Wood46 | Laboratory strain | n.a. |
| *Staphylococcus aureus* Newman | Laboratory strain | n.a. |
| *Staphylococcus epidermidis* KV103 | Clinical isolate* | n.a. |
| *Streptococcus agalactiae* COH-1 | Clinical isolate* | n.a. |

*All clinical isolates were obtained from the clinical Medical Microbiology department at the University Medical Center Utrecht.

[†]Detection of O-Ag was limited, but not absent. n.a. means O-Ag expression is not applicable, because these are Gram-positive bacteria that inherently do not express LPS.

labelled with NHS-Alexa Fluor 555 (AF555, Thermo Fisher Scientific) as done previously (*Heesterbeek et al., 2019*). The concentrations of MAC components in 100% serum are ~375 nM C5, 550 nM C6, 600 nM C7, 350 nM C8, and 900 nM C9.

## Bacterial growth

Bacterial strains that were used in this study are shown in *Table 1*. CGSC7740 wildtype (O-Ag[neg]) and *wbbL*+ (O-Ag[pos]) were kindly provided by Benjamin Sellner (Biozentrum, University of Basel). In the *wbbL*+ strain, an IS5-element that inactivates the *wbbL* gene is removed. This restores expression of an essential rhamnose transferase wbbL that is required for the expression of O-Ag (*Doorduijn et al., 2021*; *Liu and Reeves, 1994*). CGSC7740 *wbbL*+ was constructed by replacing the IS5-element present in the *wbbL*+ gene with a *sacB-kan* cassette to select for kanamycin resistance. The *sacB-kan*

cassette was then replaced with wildtype *wbbL* without the IS5-element and selected by counter selection on sucrose.

Bacteria were plated from glycerol stocks on blood agar plates. Single colonies were picked and grown overnight at 37°C in shaking conditions (600 rpm). Gram-negative bacteria were grown in lysogeny broth (LB). *S. aureus* were grown in Todd Hewitt Broth (THB) and *S. epidermidis* in Trypticase Soy Broth. *Streptococcus agalactiae* COH-1 was grown in THB in non-shaking conditions and at 5% $CO_2$. The next day, subcultures were grown by diluting at least 1/30 and these were grown to mid-log phase (OD600 between 0.4 and 0.6). Once grown to mid-log phase, bacteria were washed by centrifugation three times (11,000 rcf for 2 min) and resuspended to OD 1.0 ($1 \times 10^9$ bacteria/ml, validated by flow cytometry for all individual bacterial strains) in RPMI (Gibco) + 0.05% human serum albumin (HSA, Sanquin).

## Complement activation and killing in serum

To activate complement on bacteria, bacteria (the amount is specified in figure legends) were incubated in 5% pooled human serum for 60 min at 37°C. For sMAC ELISAs with C6-biotin, C6-depleted serum was used supplemented with 28 nM of C6-biotin. For ELISAs and Western blotting, supernatant was collected by centrifugation of bacteria at 11,000 rcf for 2 min. Blocking of C5 conversion in serum was accomplished by adding 6 μg/ml OmCI (unless otherwise specified) with and without 6 μg/ml eculizumab.

## Convertase labelling and purified MAC formation

Bacteria were labelled with convertases in C5-depleted serum as reported previously (*Heesterbeek et al., 2019*). In short, bacteria ($5 \times 10^8$ bacteria/ml) were incubated with 10% C5-depleted serum for 30 min at 37°C, washed three times (11,000 rcf for 2 min) and resuspended in RPMI-HSA. Bacteria were counted by flow cytometry to ensure that bacterial concentrations were comparable between different strains after convertase labelling. Convertase-labelled bacteria (concentrations specified in figure legends) were next incubated with MAC components (concentrations specified in figure legends) and AP convertase components (50 nM FB and 20 nM FD) for 60 min at 37°C. Supernatant for ELISA and Western blotting was collected by pelleting bacteria by centrifugation at 11,000 rcf for 2 min.

## Bacterial viability

Bacterial viability was assessed by determining CFUs. A serial dilution was made in PBS (100-, 1,000-, 10,000-, and 100,000-fold) and plated in duplicate on LB agar plates. After overnight incubation at 37°C, colonies were counted and the corresponding concentration of CFU/ml was calculated. Survival was calculated by dividing the CFU/ml in the sample by the CFU/ml at t=0.

## Flow cytometry

To measure killing by flow cytometry, 2.5 μM of Sytox Blue Dead Cell stain (Thermo Fisher Scientific) was added during the assay to measure inner membrane (IM) damage. To measure C3b deposition by flow cytometry, bacteria ($\sim 5 \times 10^7$ bacteria/ml) were stained after labelling with convertases with 3 μg/ml AF488- or AF555-labelled mouse-anti C3b for 30 min at 4°C. Finally, bacterial samples were diluted to $\sim 1 \times 10^6$ bacteria/ml in RPMI-HSA and subsequently analyzed in a MACSquant flow cytometer (Miltenyi) for forward scatter (FSC), side scatter (SSC), Sytox, AF555, and Cy5 intensity. Flow cytometry data was analyzed in FlowJo version 10. Bacteria were gated on FSC and SSC.

## Complement activation product ELISAs

Serum supernatants were diluted (as specified in figure legends) in PBS + 0.05% Tween (PBS-T) supplemented with 1% bovine serum albumin (BSA). Next, sample dilutions were analyzed for the presence of complement activation products C3a, C5a, sMAC, and C5b6 via enzyme-linked immunosorbent assays (ELISAs) on Nunc Maxisorp ELISA plates. For C5a, a sandwich-ELISA kit was used (R&D Systems, DY2037), which includes two mouse monoclonal C5a antibodies that specifically detect C5a and not native C5.

For C3a, sMAC, and C5b6, plates were coated overnight at 4°C with 1 μg/ml of coating antibody. For C3a this was mouse monoclonal anti-C3a (Hycult), for C5b6 this was monoclonal mouse

IgG1 anti human C6 (Quidel) as used previously (*Doorduijn et al., 2020*) and for sMAC this was mouse monoclonal aE11 directed against a neo-epitope of C9 in sMAC (kindly provided by T Mollness and P Garred). Blocking was next performed with PBS-T + 4% BSA for 60 min at RT. Sample dilutions were next incubated for 60 min at RT. Primary staining was performed for C3a with 1:2000 rabbit anti-C3a (Calbiochem), for C5b6 with 1:500 dilution of goat-anti human C5 serum (Complement Technology) and for sMAC with 1 µg/ml biotinylated monoclonal anti-C7 (clone F10, described in *Zelek and Morgan, 2020*, and kindly provided by Wioleta Zelek). Samples that were prepared in C6-depleted serum supplemented with C6-biotin were directly stained with 1:5000 HRP-conjugated streptavidin to detect sMAC. Otherwise, secondary staining was performed for C3a with 1:5000 HRP-conjugated polyclonal goat antisera against rabbit IgG (Southern Biotech), for C5b6 with a 1:5000 of HRP-conjugated donkey antisera against goat IgG (H+L) (Southern Biotech) and for sMAC with 1:5000 HRP-conjugated streptavidin (Southern Biotech). Finally, fresh tetramethylbenzidine was added for development and the reaction was stopped with 4N sulfuric acid to measure OD450.

At each step for all ELISAs, 50 µl was added per well, antibodies were diluted in PBS-T + 1% BSA and incubation was done for 60 min at RT (except for coating). In between steps, wells were washed three times with PBS-T in between each step. Quantification of C3a, C5a, C5b6, and sMAC was accomplished by interpolation with a standard curve of purified C3a-desarginine, C5a (Bachem), C5b6, and sMAC/SC5b-9 (Complement Technology).

## C5b Western blots

Bacterial supernatants were collected as described above and the cell pellets were also collected. Samples were diluted 1:1 in ×2 reducing SDS sample buffer (0.1 M Tris [pH 6.8], 39% glycerol, 0.6% SDS, and bromophenol blue) supplemented with 50 mg/ml dithiothreitol (DTT) and incubated at 95°C for 5 min. Samples were run on a 4–12% Bis-Tris gradient gel (Invitrogen) for 60 min at 200 V. Proteins were next transferred with the Trans-Blot Turbo Transfer system (Bio-Rad) to 0.2 µM PVDF membranes (Bio-Rad). Initially, samples were blocked with PBS supplemented with 0.1% Tween-20 (PBS-T) and 4% dried skim milk (ELK, Campina) for 60 min at 37°C. Primary staining was performed with a 1:500 dilution (~80 µg/ml) of polyclonal goat-anti human C5 (Complement Technology) in PBS-T supplemented with 1% ELK for 60 min at 37°C. Secondary staining was performed with a 1:10,000 dilution of HRP-conjugated pooled donkey antisera against goat IgG (H+L) (Southern Biotech) in PBS-T supplemented with 1% ELK for 60 min at 37°C. In between all steps and after the final staining, membranes were washed three times with PBS-T. Finally, membranes were developed with Pierce ECL Western Blotting Substrate (Thermo Fisher Scientific) for 1 min at RT and imaged on the LAS4000 Imagequant (GE Healthcare).

## Bystander lysis assay

Human erythrocytes were collected from heparin-sulfate-treated human blood of healthy donors. Erythrocytes were washed at 1000 rcf for 5 min three times with PBS. The packed erythrocyte pellet was resuspended in Veronal buffered saline (2 mM Veronal, 145 mM NaCl, pH = 7.4) supplemented with 0.1% BSA and 2.5 mM MgCl$_2$ (VBS+) and diluted to 1% (~1 × 10$^8$ erythrocytes/ml). Erythrocytes (1×10$^8$ erythrocytes/ml) were next incubated with convertase-labelled bacteria (5×10$^8$ bacteria/ml) and 100 nM C5-C8, 300 nM C9, 50 nM FB, and 20 nM FD (unless specified otherwise in figure legends) for 60 min at 37°C. Supernatant of the reaction was collected by centrifugation (1250 rcf for 5 min) and the supernatant was next diluted 1:3 in MilliQ (MQ). Hemoglobulin release was measured by measuring the absorbance at OD405 nm. The percentage of lysed erythrocytes was calculated by setting a buffer-only control at 0% lysis and an MQ control at 100% lysis. Blocking of C5 conversion in serum was done with 15 µg/ml OmCI.

## Polymeric-C9 detection by SDS-PAGE

Reaction supernatants were resuspended and diluted 1:1 in ×2 SDS sample buffer supplemented with 50 mg/ml DTT and incubated at 95°C for 5 min. Samples were run on a 4–12% Bis-Tris gradient gel (Invitrogen) for 75 min at 200 V. Gels were imaged for 10 min with increments of 30 s on the LAS4000 Imagequant (GE Healthcare) for in-gel Cy5 fluorescence. Monomeric-C9 (mono-C9) and polymeric-C9 (poly-C9) were distinguished by size, since mono-C9 runs at 63 kDa and poly-C9 is retained in the comb of the gel.

### HisTrap isolation of sMAC with His-tagged C6

*E. coli* bacteria ($5\times10^8$ bacteria/ml) were incubated in 10% C6-depleted serum supplemented with 50 nM His-tagged C6 in a total volume of 1 ml for 60 min at 37°C. Serum supernatant was collected as described above and incubated with a pellet of 900 µl Dynabeads His-Tag Isolation & Pulldown (Invitrogen) equilibrated in wash buffer (50 mM Tris, 300 mM NaCl, 10 mM imidazole, pH 7.8) on a tube rotator for 90 min at 4°C. Beads were separated using a magnet to collect the bead supernatant. Beads were washed three times in wash buffer and next His-tagged proteins were eluted with elution buffer (50 mM Tris, 300 mM NaCl, 300 mM imidazole, pH 7.8) on a tube rotator for 30 min at 4°C. Beads were separated using a magnet to collect the eluate. The eluate was finally filtered through a 0.22 µm filter and concentrated in a 100 kDa Amicon tube. sMAC was next separated from free left-over proteins by SEC on a Superose 6 Increase column with PBS. Fifty µl fractions were collected and used for subsequent analyses.

### BN-PAGE and Western blot

Samples were diluted 1:1 in ×2 NativePAGE sample buffer (Invitrogen). SDS-PAGE samples were run on a 4–12% Bis-Tris gradient gel (Invitrogen) for 75 min at 200 V. Samples were run on a NativePAGE 3–12% Bis-Tris gradient gel (Invitrogen) for 3 hr at 150 V, after which the gel was destained overnight with demineralized water. Proteins were transferred to 0.2 µM PVDF membranes with the Trans-Blot Turbo Transfer system (Bio-Rad). Membranes were blocked in PBS/0.1% Tween (PBS-T) with 4% ELK (Campina) for 45 min at 37°C. Primary detection antibodies (polyclonal goat anti-human C9 from Complement Technology) were diluted 1:500 in PBS-T/1% ELK and incubated for 45 min at 37°C. Secondary detection antibody (HRP-conjugated donkey anti-goat IgG from Southern Biotech) was diluted 1:10,000 in PBS-T/1% ELK and incubated for 45 min at 4°C. In between each step, membranes were washed three times with PBS-T. The staining was developed with Pierce ECL Western Blotting Substrate (Thermo Fisher Scientific) for 1 min at RT. Images were made using the LAS4000 Imagequant (GE Healthcare).

### SEC of activated and nonactivated serum samples

Two-hundred µl *S. aureus* Wood46 ($1.5\times10^9$ bacteria/ml) in PBS was pelleted and resuspended in 250 µl serum for 3 hr at 37°C while shaking. Bacteria were spun down at 11,000 rcf rpm for 3 min. The supernatant was collected and the centrifugation step repeated to remove remaining bacteria. The supernatant was then kept on ice and filtered through a 0.22 µm filter. SEC separation of serum samples was done using an Agilent 1290 Infinity HPLC system (Agilent Technologies) consisting of a vacuum degasser, refrigerated autosampler with a 100 µl injector loop, binary pump, thermostated two-column compartment, auto collection fraction module, and multi-wavelength detector. The dual-column set-up, comprising a tandem Yarra 4000-Yarra 3000 (SEC-4000, 300×7.8 mm ID, 3 µm, 500 Å; SEC-3000, 300×7.8 mm ID, 3 µm, 290 Å) two-stage set-up. Both columns were purchased from Phenomenex. The columns were cooled to 17°C while the other bays were chilled to 4°C to minimize sample degradation. The mobile phase buffer consisted of 150 mM ammonium acetate in water and filtered using a 0.22 µm disposable membrane cartridge (Millipore) before use. Approximately 1.25 mg of serum protein (activated and nonactivated fresh serum) was injected per run. The proteins were eluted using isocratic flow within 60 min, and the flow rate was set to 500 µl/min. In total, 74 fractions were collected within a 20–42 time window using an automated fraction collector. The chromatograms were monitored at 280 nm.

### Trypsin digestion of SEC fractions

We used bottom-up LC-MS/MS analysis to determine SEC elution profile serum proteins, isolated sMAC, and commercial sMAC. The fractions were introduced into the digestion buffer containing 100 mM Tris-HCl (pH 8.5), 1% w/v sodium deoxycholate (SDC), 5 mM Tris (2-carboxyethyl) phosphine hydrochloride, and 30 mM chloroacetamide. Proteins were digested overnight with trypsin at an enzyme-to-protein ratio of 1:100 (w/w) at 37°C. After, the SDC was precipitated by bringing the sample to 1% trifluoroacetic acid. The supernatant was collected for subsequent desalting by an Oasis µElution HLB 96-well plate (Waters) positioned on a vacuum manifold. The desalted proteolytic digest was dried with a SpeedVac apparatus and stored at –20°C. Prior to LC-MS/MS analysis, the sample was reconstituted in 2% formic acid (FA).

### LC-MS/MS analysis of isolated sMAC SEC fractions

The digested SEC fractions of isolated and commercial sMAC were analyzed using an Ultimate 3000 system (Thermo Fisher Scientific) coupled online to an Orbitrap Fusion (Thermo Fisher Scientific) controlled by Thermo Scientific Xcalibur software. First, peptides were trapped using a 0.3×5 mm PepMap-100 C18 pre-column (Thermo Fisher Scientific) of 5 µm particle size and 100 Å pore size prior to separation on an analytical column (50 cm of length, 75 µm inner diameter; packed in-house with Poroshell 120 EC-C18, 2.7 µm). Trapping of peptides was performed for 1 min in 9% solvent A (0.1% FA) at a flow rate of 0.03 ml/min. The peptides were subsequently separated by a 55 min gradient as follows: 9–13% solvent B (80% acetonitrile/0.1% FA) in 1 min, 13–44% B in 37 min, 44–99% B in 3 min, 99% B for 4 min, 99–9% B in 1 min, and finally 9% B for 8 min. The flow was 300 nl/min. The mass spectrometer was operated in a data-dependent mode. Full-scan MS spectra from 375 to 1600 Th were acquired in the Orbitrap at a resolution of 60,000 with standard automatic gain control (AGC) target and auto maximum injection time. Cycle time for $MS^2$ fragmentation scans was set to 1 s. Only peptides with charge states 1–6 were fragmented, and dynamic exclusion properties were set to n=1, for a duration of 10 s. Fragmentation was performed using HCD collision energy of 28% in the ion trap and acquired in the Orbitrap at a resolution of 15,000 and standard AGC target with an isolation window of 1.4 Th and maximum injection time mode set to auto.

### LC-MS/MS analysis of SEC serum fractions

The 74 digested SEC fractions of activated or nonactivated serum were analyzed by LC-MS/MS. Separation of digested protein samples was performed on an Agilent 1290 Infinity HPLC system (Agilent Technologies). Samples were loaded on a 100 µm×20 mm trap column (in-house packed with ReproSil Pur C18-AQ, 3 µm) (Dr Maisch GmbH, Ammerbuch-Entringen, Germany) coupled to a 50 µm×500 mm analytical column (in-house packed with Poroshell 120 EC-C18, 2.7 µm) (Agilent Technologies, Amstelveen). Ten µL of digest from each SEC fraction was used and the amount ~0.1 µg of peptides was loaded on the LC column. The LC-MS/MS run time was set to 60 min with a 300 nL/min flow rate. Mobile phases A (water/0.1% FA) and B (80% acetonitrile/0.1% FA) were used for 66 min gradient elution: 13–44% B for 35 min and 44–100% B over 8 min. Samples were analyzed on a Thermo Fisher Scientific Q Exactive HF quadrupole-Orbitrap instrument (Thermo Fisher Scientific). Nano-electrospray ionization was achieved using a coated fused silica emitter (New Objective) biased to 2 kV. The mass spectrometer was operated in positive ion mode, and the spectra were acquired in the data-dependent acquisition mode. Full MS scans were acquired with 60,000 resolution (at 200 m/z) and at a scan mass range of 375–1600 m/z. The AGC target was set to $3×10^6$ with a maximum injection time of 20 ms. Data-dependent MS/MS (dd-MS/MS) scan was acquired at 30,000 resolution (at 200 m/z) and with a mass range of 200–2000 m/z. AGC target was set to $1×10^5$ with a maximum injection time defined at 50 ms. One µscan was acquired in both full MS and dd-MS/MS scans. The data-dependent method was set to isolation and fragmentation of the 12 most intense peaks defined in a full MS scan. Parameters for isolation/fragmentation of selected ion peaks were set as follows: isolation width = 1.4 Th, HCD normalized collision energy (NCE)=27%.

### LC-MS/MS data analysis

The LC-MS/MS data were searched against UniProtKB/Swiss-Prot human proteome sequence database with MaxQuant software (version 1.5.3.30 or 2.0.3.0). For label-free quantification, iBAQ values were selected as output. For profiling of sMAC components in serum, each fraction's iBAQ values were extracted and normalized to the highest intensity.

### Data analysis and statistical testing

Unless stated otherwise, graphs are comprised of at least three biological replicates. Statistical analyses were performed in GraphPad Prism 8 and are further specified in the figure legends.

## Acknowledgements

The authors would like to acknowledge Wioleta Zelek and Paul Morgan for providing antibodies for the sMAC ELISAs, Piet Aerts for help with the silver staining, Benjamin Sellner and Urs Jenal for providing the CGSC7740 wildtype and *wbbL+* strain and Remy Muts for critical reading of the

manuscript. Financial disclosure statement: This work was funded by an ERC Starting grant (639209-ComBact, to SHMR), the Utrecht University Molecular immunology HUB (eSTIMATE), the Aspasia grant (Dutch Research Council NWO, to SHMR), the Netherlands Organization for Scientific Research (NWO) funding the Netherlands Proteomics Centre through the X-omics Road Map program (project 184.034.019, to AJRH) and fellowship support from the Independent Research Fund Denmark (Project 9036-00007B, to MVL). The funders had no role in study design, data collection and analysis, decision to publish or preparation of the manuscript.

## Additional information

### Funding

| Funder | Grant reference number | Author |
|---|---|---|
| European Research Council | (639209-ComBact | Suzan HM Rooijakkers |
| Nederlandse Organisatie voor Wetenschappelijk Onderzoek | Aspasia | Suzan HM Rooijakkers |
| Utrecht Molecular Immunology HUB | eSTIMATE | Suzan HM Rooijakkers |
| Netherlands Proteomics Centre | 184.034.019 | Albert JR Heck |
| Independent Research Fund Denmark | 9036-00007B | Marie V Lukassen |

The funders had no role in study design, data collection and interpretation, or the decision to submit the work for publication.

### Author contributions

Dennis J Doorduijn, Conceptualization, Data curation, Formal analysis, Investigation, Visualization, Methodology, Writing – original draft, Project administration, Writing – review and editing; Marie V Lukassen, Formal analysis, Funding acquisition, Investigation, Visualization, Methodology, Writing – review and editing; Marije FL van 't Wout, Validation, Investigation, Visualization, Writing – review and editing; Vojtech Franc, Formal analysis, Investigation, Visualization, Methodology, Writing – review and editing; Maartje Ruyken, Validation, Investigation; Bart W Bardoel, Conceptualization, Supervision, Methodology, Writing – original draft, Writing – review and editing; Albert JR Heck, Supervision, Funding acquisition, Writing – review and editing; Suzan HM Rooijakkers, Conceptualization, Supervision, Funding acquisition, Writing – original draft, Writing – review and editing

### Author ORCIDs

Dennis J Doorduijn http://orcid.org/0000-0002-1679-5739
Marie V Lukassen http://orcid.org/0000-0003-3237-2696
Bart W Bardoel http://orcid.org/0000-0001-6450-277X
Albert JR Heck http://orcid.org/0000-0002-2405-4404
Suzan HM Rooijakkers http://orcid.org/0000-0003-4102-0377

### Decision letter and Author response

Decision letter https://doi.org/10.7554/eLife.77503.sa1
Author response https://doi.org/10.7554/eLife.77503.sa2

## Additional files

### Supplementary files

• Transparent reporting form

## Data availability

All relevant data supporting the findings of this manuscript have been added in the main manuscript and supplemental information. Supporting source data (for figures 1 - 6a and supplements) have been uploaded to Dryad Digital Repository at https://doi.org/10.5061/dryad.g4f4qrfsd. The MS data (in figure 6, figure 6 supplement and figure 7) have been deposited to the ProteomeXchange partner MassIVE database and assigned the identifier MSV000088560 available at https://doi.org/10.25345/C5QW00.

The following datasets were generated:

| Author(s) | Year | Dataset title | Dataset URL | Database and Identifier |
|---|---|---|---|---|
| Doorduijn D | 2022 | Source data for: Soluble MAC is primarily released from MAC-resistant bacteria that potently convert complement component C5 | https://doi.org/10.5061/dryad.g4f4qrfsd | Dryad Digital Repository, 10.5061/dryad.g4f4qrfsd |
| Heck AJR | 2022 | Bottom-up MS of soluble MAC released from bacteria | https://doi.org/10.25345/C5QW00 | MassIVE MSV000088560, 10.25345/C5QW00 |

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
