## [Editor Report]

This manuscript describes in detail the strategies employed by certain bacteria to defend against lytic attack by the membrane attack complex (MAC) of complement. The major new finding is that during complement activation, these MAC-resistant bacteria are able to process and release considerable amounts of C5a as well as large amounts of a soluble form of the MAC (C5b-9) that has less C9 than the active form that promotes bacterial cell lysis.

---

## [Decision Letter]

**Decision letter after peer review:**

Thank you for submitting your article "Soluble MAC is primarily released from MAC-resistant bacteria that potently convert complement component C5" for consideration by *eLife*. Your article has been reviewed by 2 peer reviewers, and the evaluation has been overseen by a Reviewing Editor and Bavesh Kana as the Senior Editor. The following individual involved in review of your submission has agreed to reveal their identity: Ronald P. Taylor (Reviewer #2).

Essential revisions:

Overall this is a good paper that needs at a minimum fixing of the edits as suggested by both reviewers. However, the inclusion of either mechanistic data to explain why sMAC production is seen in both gram positive and negative bacteria or, alternatively, clinical correlation data looking at samples whereby there is correlation with sMAC production and the infection of sMAC-producing bacteria would substantially improve the paper.

*Reviewer #1 (Recommendations for the authors):*

Overall it is a nice paper. The intro may be too short to set up the variety of different topics that are addressed. You may be better served by pulling up elements from the very long discussion into the intro and text to help better understand the context in the various investigations that were done.

1) The O-antigen is claimed to be responsible for sMAC generation; this section needs to be reconciled with the data on gram positive bacteria. Perhaps moving part of the discussion to help put into context the mechanism of action for O-Ag would be helpful. For example lines 271-272 seem to state the MOA is unclear but then in the discussion a hypothesis is put forward.

2) In lines 366-367 there is conjecture that this may be used as a biomarker. It would be great if data from positive blood cultures could be correlated with these findings to provide clinical context.

*Reviewer #2 (Recommendations for the authors):*

Certain aspects of the presentation should be improved as noted below. However, the experiments and results constitute a complete package and there does not appear to be any experimental work needed.

Suggested changes

Line 61: "made clear" in place of "evidenced"

Line 74: Fix the grammar. Perhaps "strains released more sMAC and activated more C5 than"

Line 159: It is not clear from the figure that there was the "increase (in) C5 conversion". Can this be clarified with arrows in the figure?

Line 197: Change to "increased in the plasma of patients who suffer"

Line 223: "reported" in place of "described"

In the Discussion section, it would be very helpful if each figure and table is cited at least once.

Line 269: Expands" in place of "extends"

Line 285: Should be "for MAC-resistant"

Line 367: Should be "could also enhance our understanding of the role of complement activation in the pathogenesis of bacterial infections."

General comment: Use "centrifugation" instead of "spinning down' or "spun down."

Line 419: "accomplished by adding" in place of done"

Line 422: "reported" in place of "validated"

Line 439 Define IM

Line 453: should be "detect C5a"

Line 459: Please clarify "bound"

Line 473 : should be " sMAC was accomplished"

Line 493: should be "5 minutes three times"

Line 495: 1% corresponds to 1X 108 erythrocytes per ml (4% is wrong)

Line 502: Define MQ

Line 529: What is "demi"?

Line 551: Define AMAC

Line 587: should be "fractions of activated"

Line 595: Define ACN

Line 601: Define AGC

Figure 2b: Were statistical tests used?

Line 850: Should be: "C9 leads to release"

Figure 6d should be recast as a table

Figure 7: Colored arrows with labels should be used to better identify the curves.

Figure 6, supplement 1, line 96: should be 'are represented for"

---

## [Author Response]

Essential revisions:Overall this is a good paper that needs at a minimum fixing of the edits as suggested by both reviewers. However, the inclusion of either mechanistic data to explain why sMAC production is seen in both gram positive and negative bacteria or, alternatively, clinical correlation data looking at samples whereby there is correlation with sMAC production and the infection of sMAC-producing bacteria would substantially improve the paper.

We appreciate the positive response to our manuscript. We have now fixed the edits suggested by the reviewers, including a better explanation of the mechanism behind sMAC generation. The referees’ comments made us realize that we did not clearly explain why sMAC is generated for both Gram-positive and -negative bacteria. Our data illustrate that sMAC is primarily formed by bacteria that are resistant to MAC-mediated lysis. For Gram-negative bacteria, we show that sMAC release is associated with the expression of LPS O-Antigen in the bacterial outer membrane, which is one of the key new insights described in this paper. For Gram-positive bacteria, this is presumably because of the thick peptidoglycan layer in the bacterial cell envelope, which is also responsible for their intrinsic MAC-resistance. We therefore hypothesize that cell envelope constituents, even though they differ between Gram-negative and -positive bacteria, are involved in sMAC generation.

Several modifications were introduced to make the mechanistic link between MAC resistance and sMAC formation more explicit:

-As suggested by referee #1, we moved data showing sMAC release on Gram-positive bacteria up in the manuscript (first Figure 5, now Figure 2). This allows us to show sMAC formation by Gram-positives and explain more early in the Results section how the cell envelope of Gram-positive bacteria differs from the cell envelope of Gram-negative bacteria and how this is related to MAC-resistance (line 126-133).

-We have also adapted our discussion to explain that we hypothesize that cell envelope constituents, in the case of Gram-negatives LPS O-Antigen or for Gram-positives the thick peptidoglycan, are involved in both MAC-resistance and sMAC generation (lines 302-306)

For the suggested clinical correlation data, one would need plasma samples of a cohort of infected patients that are stored in such a way that sMAC formation can be reliably assessed (without in vitro complement activation after blood collection), with suitable controls. Since we have no such samples available, this would require set-up of a well-controlled prospective clinical trial, which we feel is outside the scope of this paper. In line with *eLife*’s policy on revisions in response to COVID-19, (https://elifesciences.org/articles/57162) we have therefore decided to slightly limit our claims and mention that such a future study would be necessary to correlate sMAC generation and MAC-resistance in a clinical setting (line 404-406).

Reviewer #1 (Recommendations for the authors):Overall it is a nice paper. The intro may be too short to set up the variety of different topics that are addressed. You may be better served by pulling up elements from the very long discussion into the intro and text to help better understand the context in the various investigations that were done.

We agree that the discussion is relatively long. We have omitted a part of the discussion that goes into depth on C5 convertases (line 315-322), as we agree that this would require more introduction but is out of scope of the main topic of this manuscript.

1) The O-antigen is claimed to be responsible for sMAC generation; this section needs to be reconciled with the data on gram positive bacteria. Perhaps moving part of the discussion to help put into context the mechanism of action for O-Ag would be helpful. For example lines 271-272 seem to state the MOA is unclear but then in the discussion a hypothesis is put forward.

We now emphasize in the beginning of the discussion (lines 299-306) that cell envelope composition of bacteria in general, rather than just the LPS O-Ag, could be responsible for the difference in sMAC release observed in our study. Our data illustrate that sMAC is primarily formed by bacteria that are resistant to MAC-mediated lysis. For Gram-positive bacteria, this is presumably because of the thick peptidoglycan layer in the bacterial cell envelope, which is also responsible for their intrinsic MAC-resistance. For Gram-negative bacteria, we show that sMAC release is associated with the expression of LPS O-Antigen in the bacterial outer membrane, which is one of the key new insights described in this paper. This is why we discuss the LPS O-Antigen in more depth than the potential membrane structures that could cause this in Gram-positives, because we have more direct evidence that links sMAC generation to the expression of LPS O-Antigen in MAC-resistant Gram-negative bacteria.

2) In lines 366-367 there is conjecture that this may be used as a biomarker. It would be great if data from positive blood cultures could be correlated with these findings to provide clinical context.

For the reader’s understanding, we have adapted our phrasing here (lines 402-404) to ensure that the reader understands that sMAC is already used as a biomarker in bacterial infections (reference 16, 17 and 18). Our present study focuses on how this biomarker is formed and what it represents. We also now mention in the discussion that a future study would be necessary to correlate sMAC generation and MAC-resistance in a clinical setting (line 404-406). Such a prospective clinical trial would entail a study on its own, as sMAC will have to be monitored more closely during multiple stages of infection. Patients would have to be specifically included for this study, as samples would also have to be collected in a way that prevents sMAC generation after blood collection.

Reviewer #2 (Recommendations for the authors):Certain aspects of the presentation should be improved as noted below. However, the experiments and results constitute a complete package and there does not appear to be any experimental work needed.Suggested changesLine 61: "made clear" in place of "evidenced"

This has been adapted in the revised manuscript.

Line 74: Fix the grammar. Perhaps "strains released more sMAC and activated more C5 than"

This has been adapted in the revised manuscript.

Line 159: It is not clear from the figure that there was the "increase (in) C5 conversion". Can this be clarified with arrows in the figure?

This has been adapted in the revised manuscript.

Line 197: Change to "increased in the plasma of patients who suffer"

This has been adapted in the revised manuscript.

Line 223: "reported" in place of "described"

This has been adapted in the revised manuscript.

In the Discussion section, it would be very helpful if each figure and table is cited at least once.

This has been adapted in the revised manuscript.

Line 269: Expands" in place of "extends"

This has been adapted in the revised manuscript.

Line 285: Should be "for MAC-resistant"

This has been adapted in the revised manuscript.

Line 367: Should be "could also enhance our understanding of the role of complement activation in the pathogenesis of bacterial infections."

This has been adapted in the revised manuscript.

General comment: Use "centrifugation" instead of "spinning down' or "spun down."

This has been adapted in multiple sections in the revised manuscript.

Line 419: "accomplished by adding" in place of done"

This has been adapted in the revised manuscript.

Line 422: "reported" in place of "validated"

This has been adapted in the revised manuscript.

Line 439 Define IM

This has been adapted in the revised manuscript.

Line 453: should be "detect C5a"

This has been adapted in the revised manuscript.

Line 459: Please clarify "bound"

This has been adapted in the revised manuscript.

Line 473 : should be " sMAC was accomplished"

This has been adapted in the revised manuscript.

Line 493: should be "5 minutes three times"

This has been adapted in the revised manuscript.

Line 495: 1% corresponds to 1X 108 erythrocytes per ml (4% is wrong)

This has been adapted in the revised manuscript.

Line 502: Define MQ

MQ is defined in line 544.

Line 529: What is "demi"?

This has been adapted in the revised manuscript.

Line 551: Define AMAC

This has been defined (ammonium acetate) in the revised manuscript.

Line 587: should be "fractions of activated"

This has been adapted in the revised manuscript.

Line 595: Define ACN

ACN has been replaced with its definition acetronitrile in the revised manuscript (also in line 622).

Line 601: Define AGC

This has been defined in line 626 (automatic gain control) in the revised manuscript.

Figure 2b: Were statistical tests used?

A statistical test has been performed to compare MAC-sensitive strains versus MAC-resistant strains. This has been described in the figure legend (line 940-942) and the relevant p-value has been added to the figure.

Line 850: Should be: "C9 leads to release"

This has been adapted in the revised manuscript.

Figure 6d should be recast as a table

The mean ratio values of the relevant components that are discussed in the Results section have been added above the dot plots and described in the figure legend (line 1033-1034). The tables with all individual samples are added on Dryad and can always be viewed in case the reader wants to.

Figure 7: Colored arrows with labels should be used to better identify the curves.

Colored arrows have been added on top to indicate where the peaks of individual components are located in both samples.

Figure 6, supplement 1, line 96: should be 'are represented for"

This has been adapted in the revised manuscript.